# Ultrastretchable and superior healable supercapacitors based on a double cross-linked hydrogel electrolyte

Huili Li[1], Tian Lv[1], Huanhuan Sun[1], Guiju Qian[1], Ning Li[1], Yao Yao[1] & Tao Chen [1]

Due to inherently poor healable and stretchable features, the most explored polyvinyl alcohol-based gel electrolytes cannot well meet the requirements of stretchable, healable and multifunctional supercapacitors. Here, we report a hydrogel of a copolymer cross-linked by double linkers of Laponite (synthetic hectorite-type clay) and graphene oxide. The resultant hydrogel shows high mechanical stretchability, excellent ionic conductivity, and superior healable performance. Along with designing wrinkled-structure electrodes, supercapacitors fabricated by using this hydrogel as a gel electrolyte not only exhibit ultrahigh mechanical stretchability of 1000%, but also achieve repeated healable performance under treatments of both infrared light irradiation and heating. More significantly, a broken/healed super-capacitor also possesses an ultrahigh stretchability up to 900% with slight performance decay. This hydrogel electrolyte could be easily functionalized by introducing other functional components, and extended for use in other portable and wearable energy related devices with multifunction.

[1] Shanghai Key Lab of Chemical Assessment and Sustainability, School of Chemical Science and Engineering, and Institute of Advanced Study, Tongji University, 200092, Shanghai, China. Correspondence and requests for materials should be addressed to T.C. (email: tchen@tongji.edu.cn)

The newly emerging healable and stretchable electronics (e.g., wearable electronics[1] and smart electronic textiles[2]) have recently attracted extensive attention due to their excellent flexible, smart, and anti-deformation properties. To well satisfy the power requirements of aforementioned electronics, it is urgent to develop healable and stretchable energy storage devices having a favorable match with them. Due to their high power density, long cycling durability and simple fabrication process, flexible wearable supercapacitor devices with either healable[3] or stretchable[4] properties have been investigated intensely through designing novel structure electrodes[5,6] and electrolytes[7,8]. However, the reported stretchable supercapacitors often possess a relatively low stretchability (typically <400%) but without healable capability[9,10], while the healable supercapacitors had a limited repeated healing stability (<10 broken/healed cycles) and poor stretchability[8,11]. Few stretchable and healable supercapacitors have been demonstrated, but often suffer from poor electrochemical performance. Investigations have not been focused whether the stretchable properties could be well maintained after breaking/healing processes.

Generally, the healable and/or stretchable properties of supercapacitors are provided by elastic substrates[12–15] or electrolytes[16,17]. For electrolytes, the most used polyvinyl alcohol (PVA)-based electrolytes possess neither intrinsically healable nor stretchable properties, often resulting in supercapacitor devices with poor healable performance and limited stretchability[11,12,18]. In this regard, cross-linked polymer hydrogels that can be easily enabled with high ionic conductivity, healable, and stretchable properties represent very promising candidates for use as gel electrolytes for multifunctional supercapacitors. However, relatively little research work has been performed to address this issue[19–21]. Here, we report a nanocomposite hydrogel of poly(2-acrylamido-2-methylpropane sulfonic acid-co-N,N-dimethyla-crylamide) (poly(AMPS-co-DMAAm)) cross-linked by synthetic hectorite-type clay (also called as Laponite) and graphene oxide (GO). The presence of GO, with excellent mechanical and electrical properties[22], provides the resultant hydrogels with high mechanical performance (tensile strength of 34 kPa and stretchability of 1173%) and excellent ionic conductivity. The abundant functional groups (-COOH, -OH, and $Mg^{2+}$) in Laponite and GO can greatly facilitate a cross-linking reaction with groups (-$CONH_2$) in polymer chains at the broken interface of hydrogel, enabling outstanding repeatable healable performance under conditions of either heating or infrared light. All the above features make this novel hydrogel attractive for use as a gel electrolyte for healable and/or stretchable supercapacitors. Through optimizing the electrode structure and device fabrication procedure, the resultant two-electrode-based supercapacitor can not only maintain its original electrochemical performance under a high strain of 1000%, but also can withstand repeated stretching to the strain of 300% for 2000 cycles with slight performance decay (2%). Meanwhile, these supercapacitor devices can also recover their original performance after being repeatedly cut and healed by either heating or infrared light treatment, suggesting excellent healable properties. More importantly, the healed supercapacitor device can also be stretched to a strain of 900% with only 15% of performance decrease, indicating outstanding mechanically healable property. Both the stretchability and healable performances of our supercapacitors are much higher than those of other devices reported previously[11,19,23], to the best of our knowledge.

## Results

### Fabrication and properties of nanocomposite hydrogels.

The desired poly(AMPS-co-DMAAm)/Laponite/GO nanocomposite hydrogels (Figs. 1a, b) were synthesized by in-situ co-polymerization of the monomers AMPS and DMAAm with GO (Supplementary Figure 1) and Laponite (Supplementary Figure 2) served as collaborative cross-linking agents. The obtained hydrogel showed a typical three-dimensional network structure (Supplementary Figure 3a) with cross-linkers dispersed uniformly (Supplementary Figure 3b), which indicated the nanocomposite hydrogel having a uniform structure. From the fourier transform infrared (FTIR) spectra (Supplementary Figure 4), the formed hydrogel appeared the Si-O stretching and Si-O-Si stretching bands at 1037 $cm^{-1}$ and 622 $cm^{-1}$, respectively, which had an obvious shift compared with bare Laponite (1041 $cm^{-1}$ and 646 $cm^{-1}$). All of the C–O stretching at 1360 $cm^{-1}$, C-O-C stretching band at 1037 $cm^{-1}$ and the C=O stretching band at 1620 $cm^{-1}$ in the hydrogel showed a dramatic shift, compared with that of GO (1375 $cm^{-1}$, 1041 $cm^{-1}$, and 1720 $cm^{-1}$), which can confirm that crosslinking by hydrogen bonds was formed between GO and polymer chains. The stretchable and healable properties of hydrogels have been systematically adjusted by changing the contents of monomers and cross-linkers during co-polymerization process (Supplementary Figure 5-7). The results showed that the polymer hydrogel can achieve a favorable balance among mechanical strength, healable efficiency, and ionic conductivity, as the mass percentages of AMPS, DMAAm, Laponite, and GO in precursor solution for polymerization were 4.8 wt%, 12.6 wt%, 2 wt%, and 0.1 wt%, respectively.

The healable capability of hydrogels was investigated by comparing their mechanical properties and electrical conductivities before and after a broken/healed process. As shown in Fig. 1c, a broken hydrogel completely recovered its strain of 1169% along with an enhanced tensile strength under irradiation of 808 nm infrared light within 20 min. The rapidly and highly healable performance of the nanocomposite hydrogel can be ascribed to its excellent photothermal property derived from GO with high optical absorbance[24], which could be confirmed by the obvious temperature increase as the hydrogel containing GO was irradiated under infrared light (Supplementary Figure 8). The infrared light energy and the generated heating energy greatly promoted the diffusion of polymer chains and re-crosslinking reaction in the hydrogel, resulting in a rapid healing behavior and an enhanced tensile strength after a broken/healed process. Meanwhile, a broken hydrogel could also be healed to its original strain by heating at 80 ºC for 60 min (Fig. 1d). The healable property of hydrogels can be ascribed to the rapid diffusion of poly(DMAAm-co-AMPS) chains at the interface of broken place and re-interacting with Laponite and/or GO under the condition of heating or infrared light. On the other hand, re-cross-linking interaction also happened in other place of the whole hydrogel beyond the broken interface during the healing process, which provided the healed hydrogels with an enhancement of both mechanical strength (Figs. 1c, d) and storage modulus (G') (Supplementary Figure 9). Due to the presence of GO with abundant oxygen-containing functional groups[25–27], the poly(AMPS-co-DMAAm)/Laponite/GO nanocomposite hydrogels also possessed a comparable ionic conductivity (6.2–30.0 mS $cm^{-1}$, Supplementary Figure 7) with conventionally used PVA/$H_3PO_4$ electrolyte[28], which meant that these hydrogel could be used as a candidate electrolyte for multifunctional supercapacitors. More interestingly, the electrical conductivity of the obtained hydrogels achieved a healing efficiency of near 98% after >15 cycles (Fig. 1e) of broken and healed by both treatments of infrared light irradiation and heating (Supplementary Figure 10), suggesting excellent healable performance. From both scanning electron microscopy (SEM) images (Figs. 1f, g) and microscope images (Supplementary Figure 11 and 12), it can be confirmed that the broken place was completely repaired by both methods of heating and infrared light irradiating.

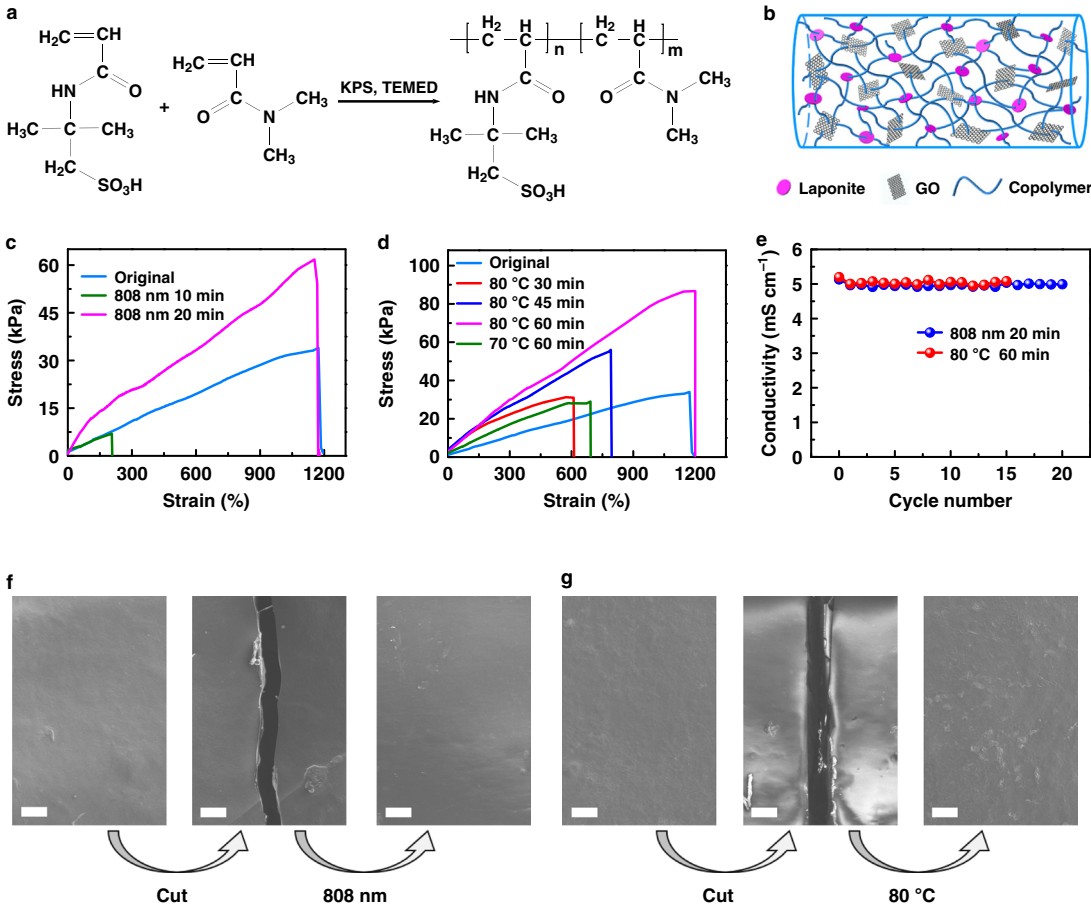

**Fig. 1** Synthesis and properties of the nanocomposite hydrogels. **a** Schematic of copolymerization of poly(AMPS-co-DMAAm) with potassium persulfate (KPS) and *N,N,N′,N′*-tetramethylethylenediamine (TEMED) served as initiator and catalyst, respectively. **b** Schematic of poly(AMPS-co-DMAAm)/Laponite/graphene oxide (GO) nanocomposite hydrogels with Laponite and GO served as cross-linkers. **c, d** Healable property of hydrogels under irradiation of 808 nm infrared light for 20 min (**c**) and heating treatment at different temperature (**d**). **e** Electrical conductivity retention of a hydrogel film after cyclic broken/healed process. **f, g** Scanning electron microscopy (SEM) images of poly(AMPS-co-DMAAm)/Laponite/GO hydrogels being healed by treatment of 808 nm infrared light irradiation for 20 min (**f**) and 80 °C for 60 min (**g**). The scale bar: 100 μm

**Electrochemical performance of the hydrogel electrolyte-based supercapacitors**. To investigate the potential application of the developed hydrogel electrolyte for energy storage, supercapacitor devices were fabricated by sandwiched one piece of hydrogel film (served as electrolyte and separator, simultaneously) with two carbon nanotube (CNT) film electrodes, and their electrochemical performance were characterized by electrochemical working station. The achieved supercapacitor devices showed typical rectangular cyclic voltammetry (CV) curves (Supplementary Figure 13a) measured at the scan rates ranging from 0.1 to as high as 3.0 V s$^{-1}$ and near triangular shape of galvanostatic charge/discharge (GCD) curves (Supplementary Figure 13b) at different current density, respectively, suggesting ideal capacitive behavior and rate performance. Significantly, the supercapacitor by using our hydrogel as gel electrolyte exhibited an over double specific capacitance compared with that of supercapacitor based on traditional PVA/LiCl gel electrolyte (Supplementary Figure 13c). In addition, the performance of supercapacitors based on the nanocomposite hydrogel electrolyte could be greatly improved from 9 mF cm$^{-2}$ (bare CNTs electrodes) to 180 mF cm$^{-2}$ (Supplementary Figure 14) as CNT/polyaniline (PANI) composites (Supplementary Figure 15) were used as electrodes. Both supercapacitor devices based bare CNTs and CNT/PANI composites not only showed an excellent cycling charge/discharge

performance (Supplementary Figure 16), but also exhibited outstanding mechanically flexible stability (Supplementary Figure 17). All the aforementioned results revealed that our newly developed nanocomposite hydrogel was an efficient candidate of gel electrolyte for multifunctional supercapacitor devices with high performance.

**Stretchable property of the hydrogel-based supercapacitors**. Benefiting from its high ionic conductivity, healable, and stretchable properties, the newly developed nanocomposite hydrogel of poly(AMPS-co-DMAAm)/Laponite/GO could be used as an ideal candidate of gel electrolyte for multifunctional supercapacitor devices. Stretchable supercapacitor devices were developed by attaching two CNT films on both sides of a pre-strained (900%) hydrogel electrolyte film, followed by releasing of the pre-strain (Fig. 2a). As expected, the formed wrinkled CNTs electrodes (Fig. 2b) became plane (Fig. 2c) as 900% tensile strain was applied on an obtained supercapacitor, and then the random CNT network turned into aligned structure (Fig. 2d) but without damage under the strain of 1000%. The unique morphological evolution of CNT electrode enabled the supercapacitor devices to maintain stable electrochemical performance during stretching process. The electrochemical measurements showed that the

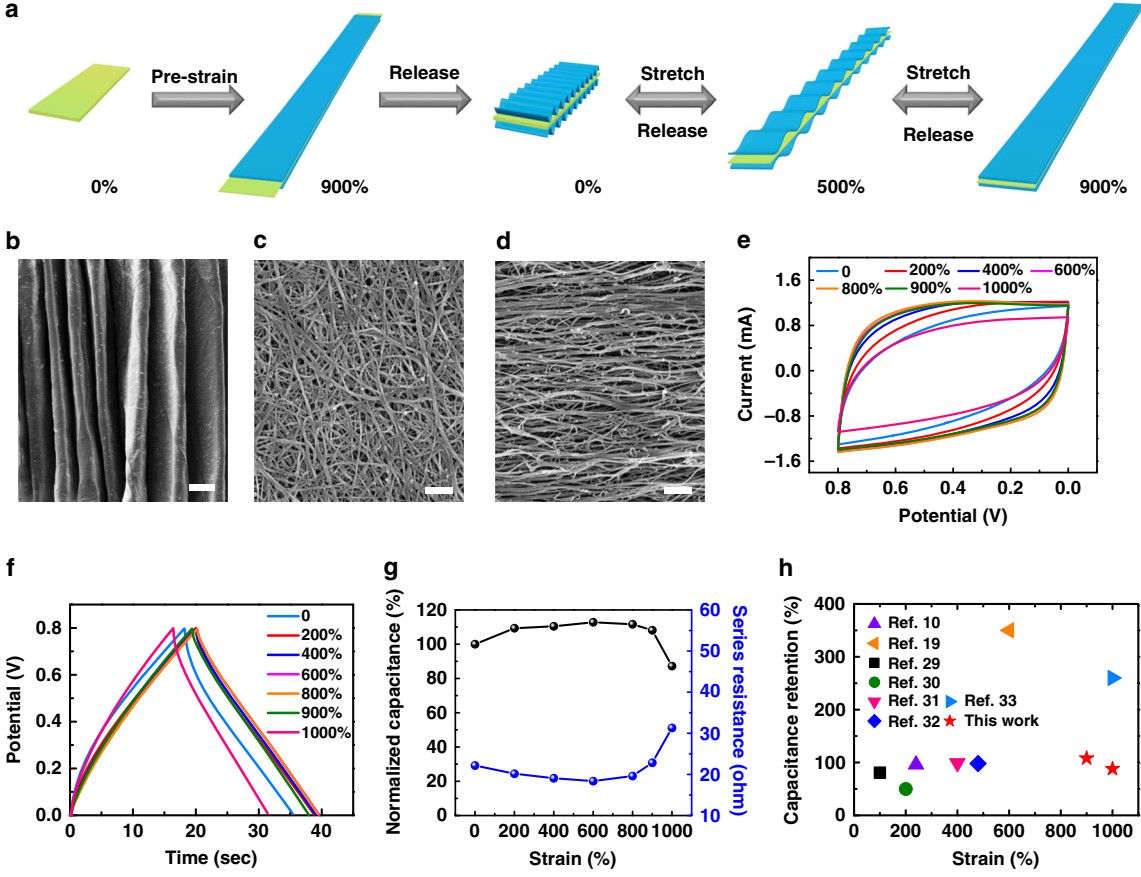

**Fig. 2** Fabrication and properties of the stretchable supercapacitor devices. **a** Schematic for fabrication of a stretchable supercapacitor by using the hydrogel as electrolyte and CNT films as electrodes through a pre-strain/release process. **b–d** Scanning electron microscopy (SEM) images of the formed wrinkled CNTs (**b**, scale bar: 200 μm) in a supercapacitor being re-stretched to strains of 900% (**c**, scale bar: 500 nm) and 1000% (**d**, scale bar: 500 nm). **e–g** Cyclic voltammetry (CV) curves (**e**), galvanostatic charge/discharge (GCD) curves (**f**) and normalized electrochemical performance (**g**) of a supercapacitor being stretched from 0 to 1000% strain. **h** Comparison of capacitance retention and stretchability of our stretchable supercapacitors with other results reported previously

supercapacitor device remained typically rectangular CV curves (Fig. 2e) and triangular GCD curves (Fig. 2f) even under the tensile strain as high as 1000%, revealing excellent retention of its capacitive behavior. As shown in Fig. 2g, the supercapacitor showed a higher capacitance than that of its original state as the tensile strains increased from 0 to 800%, which can be attributed to the decrease of series resistance (Supplementary Figure 18) derived from the increasing contact area between electrolyte and CNT electrodes, with wrinkled CNTs becoming plane (Figs. 2b, c). Then, the capacitance of supercapacitor slightly declined as the strains increased from 800% to 900%, because the contact area between electrolyte and CNTs electrodes changed very limited but the series resistance increase. As the supercapacitor was further stretched from 900% to 1000%, the series resistance sharply increased (Supplementary Figure 18), which resulted in a decrease of capacitance, but almost maintaining 87% of its original value under the strain of 1000%. Furthermore, the supercapacitor devices based on this hydrogel electrolyte possessed capacitance retention of over 98% as they were repeatedly stretched to the strains of 100%, 200%, and 300% for 2000 cycles (Supplementary Figure 19), suggesting excellent mechanically stretchable stability. To the best of our knowledge, the mechanical stretchability and cycling stability of our newly developed supercapacitors are among the best range compared with counterparts reported previously (Fig. 2h)[10,19,29–33].

**Healable performance of the hydrogel-based supercapacitors.** Besides stretchable devices, highly healable supercapacitor devices could also be realized based on this nanocomposite hydrogel electrolyte. As shown in Figs. 3a, b, GCD curves almost overlapped as a supercapacitor that was repeatedly cut almost in the same position and healed through both approaches of irradiating under infrared light of 808 nm for 10 min and heating at 80 °C for 60 min. Due to the used CNTs with excellent optical absorbance and photothermal property[34–36], the temperature of supercapacitor devices rapidly increased to over 80 °C (Supplementary Figure 20) as the infrared light was irradiated on the surface of CNTs electrode, achieving a much more rapid healing process (10 min) than that (20 min) of bare hydrogel. The supercapacitors could recover over 90% of their original capacitances (Fig. 3c) with slight increase of series resistance (Supplementary Figure 21) after broken and healed by heating and infrared irradiation for 15 and 20 cycles, respectively, which is much higher than that of other reported healable supercapacitor devices (Fig. 3d)[11–13,21,23,37,38]. The healed supercapacitor also possessed a superior cycling stability (91.88% and 92.96% capacitance retention) for 10,000 charge–discharge cycles (Supplementary Figure 22a-c), comparative with that of the as-prepared device (Supplementary Figure 22d). Furthermore, the performance of healed supercapacitor remained almost unchanged as the device was bent to any angle (Fig. 3e) and even repeatedly bent for 5000

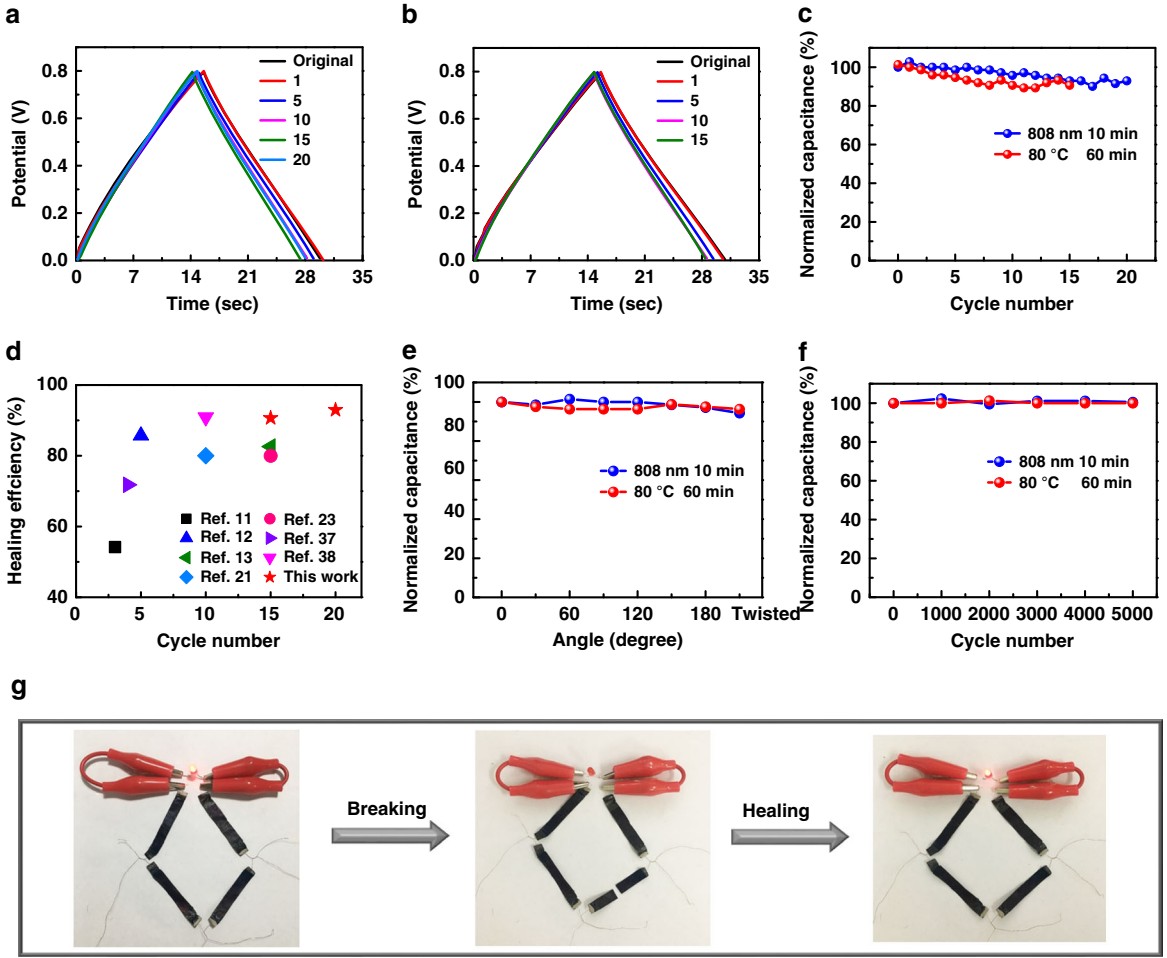

**Fig. 3** Healing performance of supercapacitors based on hydrogel electrolyte. **a**, **b** Galvanostatic charge/discharge (GCD) curves of supercapacitor devices healed by irradiation with 808 nm infrared light for 10 min (**a**) and heating at 80 °C for 60 min (**b**) for different broken/healed cycles. The numbers 1, 5, 10, 15, and 20 correspond to the number of cycles. **c** Specific capacitance retention of supercapacitor devices by broken/healed for various cycles. **d** Comparison of the healing efficiency and cycling number of our supercapacitors with previously reported results. **e**, **f** Specific capacitance retention of a healed supercapacitor under different bending angles (**e**) and after 5000 bending cycles to an angle of 120° (**f**). **g** Digital photographs of four supercapacitors connected in series to power a light-emitting diode (LED) with one device being performed for cut/healed operation

cycles (Fig. 3f), indicating predominant flexibility and mechanically bending stability. There was no difference for a supercapacitor connected in series with other three devices to power a light-emitting diode (LED, operation voltage of 1.8 V) before and after broken/healed procedure (Fig. 3g).

## Discussion

Despite the demonstration of many healable supercapacitors elsewhere[12,16,19,37,39,40], reports on healed supercapacitor devices that can recover mechanical stretchability have been limited so far. In this work, the unique wrinkled structure of CNT electrodes and the excellent healing properties (both mechanical strain and ionic conductivity) of the nanocomposite hydrogel electrolyte enabled a healed supercapacitor device with a high mechanical stretchable property (Fig. 4a). As Figs. 4b–d show the broken supercapacitor device that was healed by both heating and infrared light irradiation can withstand a tensile strain as high as 900%, very close to that of the as-prepared devices. The healed device has a very slight capacitance degradation (8%) as it is stretched to the strain of 800% and 15% at the strain of 900% (Fig. 4d) because of its limited increase of series resistance

(Supplementary Figure 23), suggesting highly efficient healing performance. Meanwhile, the healed supercapacitors maintain their electrochemical performance as they are stretched to strains of 100%, 200%, and 300% for 2000 cycles (Supplementary Figure 24 and 25), respectively, indicating good cycling stretching stability. As shown in Fig. 4e, an LED can be powered by in-series connected supercapacitors with one healed device being stretched to a strain of 800%. The ultrahigh stretchability and superior healable performance of both as-prepared and broken/healed supercapacitors can be mainly ascribed to the outstanding mechanical and healable performance of the developed hydrogel electrolyte. On the other hand, the wrinkled structure of CNT film electrodes enable the supercapacitor to maintain its electrochemical performance under high tensile strain, and also provides CNT electrodes with abundant surface to contact again during healing process, resulting in such highly stretchable and healable supercapacitor devices. It can be expected that more stretchable and healable supercapacitors could be achieved if non-healable CNT-based electrodes in this work were replaced by other healable electrode materials.

In summary, a multifunctional hydrogel polyelectrolyte was constructed by in situ co-polymerization of DMAAm and AMPS

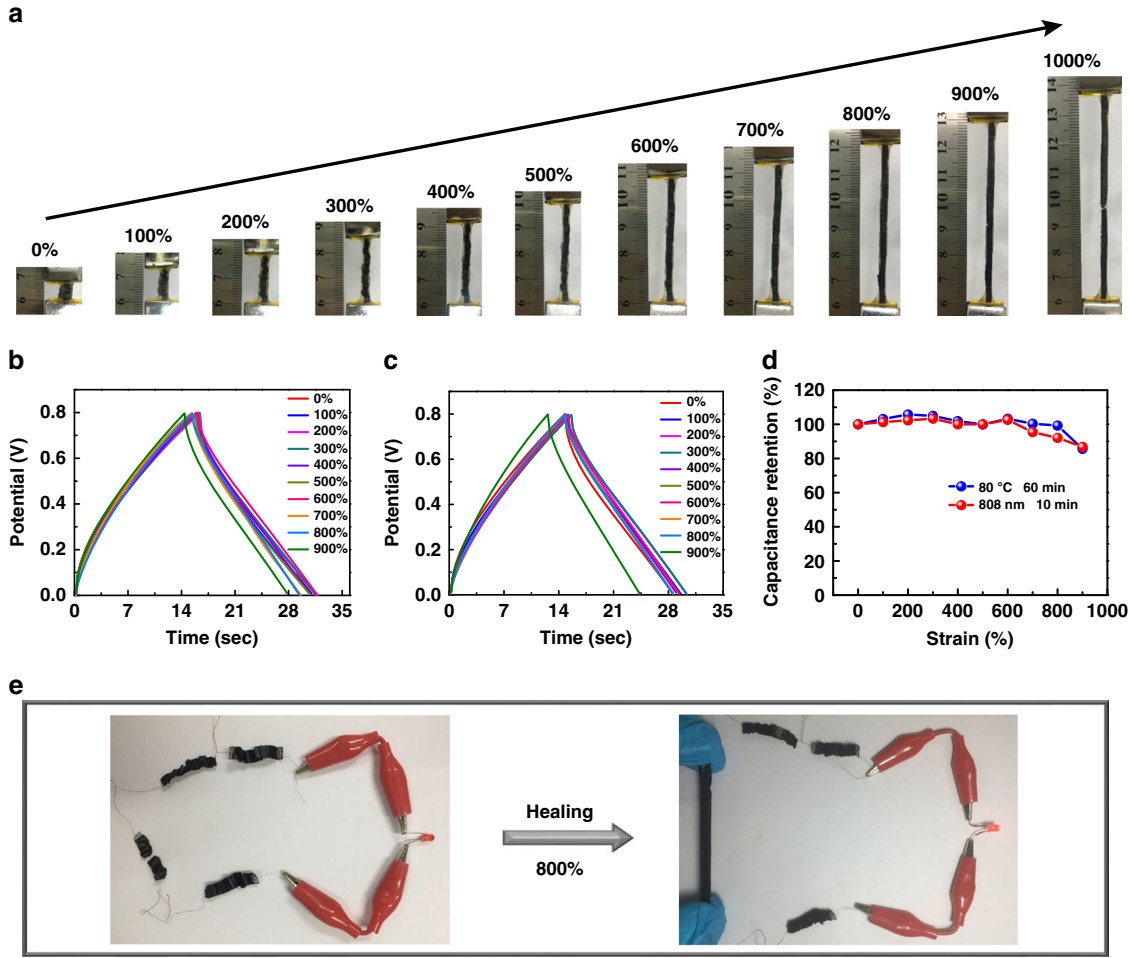

**Fig. 4** Stretchable properties of a broken/healed supercapacitor. **a** Digital photographs of a broken/healed supercapacitor being stretched from 0 to 1000%. **b**, **c** Galvanostatic charge/discharge (GCD) curves of a broken supercapacitor healed by infrared light for 10 min (**b**) and heating treatment at 80 °C for 60 min (**c**) as they were stretched from 0 to 900%. **d** Specific capacitance retention of the broken and healed supercapacitors calculated from the results of **b** and **c**. **e** Digital photographs of four supercapacitors connected in series to power a light-emitting diode (LED), in which one device was broken/healed and then stretched to a strain of 800%

with cross-linkers of GO and Laponite. The achieved hydrogel exhibited ionic conductivity that is comparable with traditional PVA electrolyte, intrinsic stretchability (1173% and 34 kPa) and excellent healable performance under either heating or infrared light treatment. By using this novel nanocomposite hydrogel as both a gel electrolyte and a separator, the developed all-solid-state supercapacitors not only exhibit superior mechanical flexibility and stretchability (up to 1000%), but also possess repeated healable performance. Furthermore, a healed supercapacitor device shows flexibility and stretchability that are comparable with the as-prepared device, which outperform other reported counterpart devices to the best of our knowledge. The demonstrated nanocomposite hydrogel may be further functionalized with ionic liquid or other functional materials, enabling interesting characteristics and potential applications in other energy storage systems.

## Methods

**Preparation of nanocomposite hydrogels**. The poly(AMPS-co-DMAAm)/Laponite/GO nanocomposite hydrogels were synthesized through an in-situ co-polymerization of the monomers AMPS and DMAAm in the presence of GO and Laponite dispersion. Typically, GO (0.32 g, 2.5 wt%) was dispersed in deionized water (7.8 mL) with continuous stirring for 20 min to achieve homogeneous dispersion, followed by ultrasonic vibration for 30 min. Then, Laponite (0.16 g) was added into the GO suspension with stirring for 15 min. AMPS (0.38 g), DMAAm (1.05 mL), and initiator potassium persulfate (KPS; 0.01 g) were

added subsequently with stirring for 15 min as one of them was added. Finally, catalyst of $N,N,N',N'$-tetramethylethylenediamine (TEMED; 10 µL) was blended along with stirring for 5 min to obtain the precursor. The precursor was poured into a home-made mold and followed by an in-situ free radical co-polymerization in ambient temperature for 24 h. The as-prepared hydrogel films (thickness ranging from 0.8 to 1.5 mm) were cut into desired shapes for characterization or use in supercapacitors.

**Preparation of supercapacitor devices**. CNT films with thickness of about 10 µm were cut into a desired size (typically 2 mm × 15 mm), with one end connected with a copper wire using silver paste for facilitating electrochemical measurement. Healable supercapacitors were assembled by directly attaching two pieces of CNT-based film electrodes on both sides of a hydrogel electrolyte film. Stretchable supercapacitors were fabricated by attaching two pieces of CNT-based film electrodes on both sides of a piece of pre-strained (900%) hydrogel electrolyte film, and followed by releasing of the pre-strain. For healing measurement of hydrogel electrolyte-based supercapacitor devices, the broken ends of non-healable CNT-based films were overlapped and pressed to adhere on the surface of hydrogel electrolyte by hand to make them contacted again. To prevent evaporating of water, the hydrogel films and supercapacitor devices were stored in a chamber with constant temperature (25 °C under infrared light and 80 °C for heating) and humidity (98%) during cyclic broken/healed processes.

**Characterization**. FTIR spectra of Laponite, GO, and dry hydrogels were conducted through an IR spectrophotometer (Alpha, Bruker) with the range of wavenumbers from 400 to 4000 cm$^{-1}$. The morphologies of the hydrogels were observed via field scanning electron microscopy (FESEM, Hitachi S-4800) at an accelerating voltage of 5 kV. Freeze-dried hydrogels were carried out

through a SCIENTZ-10N before SEM characterization. The structure of Laponite was characterized by using high-resolution transmission electron microscopy (HRTEM, JEOL-2010) operated at an accelerating voltage of 200 kV. Microscope images of hydrogels colored with a dye of fast green FCF - (purchased from Aladdin Industrial Corporation, China) were taken by fluorescence microscope (Eclipse Lv100npol, Nikon). Storage modulus and loss modulus of hydrogels were measured by rheometer (HAAKE RheoStress 6000, Thermo Fisher Scientific). Thermal mapping images of CNT film, hydrogels, and supercapacitors were taken by a thermal infrared instrument (PI 450, Optris). The electrochemical measurements were recorded using an electrochemical systems (CHI 760E, Shanghai Chenhua). The mechanical flexibility and stretchability of hydrogels and supercapacitors were evaluated by fixing them in a mechanical test machine (HY-0350, Shanghai Hengyi Co. Ltd) and recording their electrochemical performance under a given bending state or tensile strain.

## Data availability

The data that support the findings of this study are available from the corresponding author upon request.

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

## Acknowledgements

This work is supported by the National Natural Science Foundation of China (51503152, 21774094, and 51702237), Science & Technology Commission of Shanghai Municipality (14DZ2261100), Shanghai Rising-Star Program (17QA1404300), the Youth Talent Support Program at Shanghai, the Program for Professor of Special Appointment (Eastern Scholar) at Shanghai Institutions of Higher Learning, and the Fundamental Research Funds for the Central Universities (Tongji University).

## Author contributions

T.C. conceived this idea and supervised the work. H.L. contributed to most experimental work. T.L., H.S., G.Q., N.L., and Y.Y. assisted in characterization of some materials. T.C. and H.L. wrote the manuscript. All the authors assisted in experiments and provided constructive comments on the text.

## Additional information

**Competing interests:** The authors declare no competing interests.

