## [Peer Review File · Nature Communications]

Reviewers' Comments:

Reviewer #1:

Remarks to the Author:

The authors reported a hydrogel of a copolymer cross-linked by dual linkers of Laponite and graphene oxide. The unique properties and abundant functional groups of cross-linkers provide the resultant hydrogel with high mechanical stretchability, excellent ionic conductivity, and superior self-healable performance under both infrared light and heating treatments. Along with designing wrinkled structure electrodes, supercapacitors fabricated by using this hydrogel as gel electrolyte not only exhibit superior mechanical stretchability of 1000%, but also could achieve highly efficiently repeated self-healable performance with treatments by infrared irradiation and heating.

However, this manuscript should be revised in the following items before it can be accepted for publication.

1. The specific capacitance slightly decreased with the strain. The author ascribed to the increasing series resistance of devices. However, the contact area between electrode and electrolyte may be changed. The author should give the comprehensive analysis from different reasons.

2. The author should give EIS testing result. For better confirm the good performance of self-healing and stretching, some analysis about the EIS needs to be given.

3. The devices were cut in same position? The author should explain.

Reviewer #2:

Remarks to the Author:

This manuscript designed a kind of hydrogel electrolyte based on poly(AMPS-co-DMAAm) cross-linked by Laponite and graphene oxide. The resulting electrolyte showed high stretchability and healability. A supercapacitor containing the hydrogel electrolyte not only kept the original electrochemical performances at a high strain of 1000%, but also exhibited desirable healability after multiple mechanical damages. Interestingly, the healed capacitor could also be stretched to a strain of 900% with only 15% capacitance decay. The present investigation offers an alternative strategy to fabricate flexible capacitors with healability and stretchability. However, the following questions should be clarified before the manuscript can be accepted by Nature Communications.

1) Although the capacitor described by Dr. Chen exhibited healability and stretchability, the similar capacitor and the fabrication strategy had been reported by Chunyi Zhi et al (Nat. Chem. 2015, 6, 10310).

2) The healing behavior of the capacitor is attributed to its hydrogel electrolyte. However, the electrodes and current collectors of the capacitor did not show self-healing capability.

3) The mechanical recovery of the broken hydrogel electrolyte involved external stimulus such as infrared light and heat treatment. In this regard, the present hydrogel electrolyte is only healable instead of self-healable. In addition, the temperature of the spot irradiated by infrared light should be measured. Will the temperature higher than 80 °C?

Reviewer #3:

Remarks to the Author:

This manuscript reports new type of poly (AMPS-co-DMAAm)/Laponite/GO nanocomposite hydrogels

with ultra-high stretchability and self-healing properties. It is very meaningful for the development of wearable electronic devices and smart electronic textiles. However, some revisions need to be made before its publication.

1. The presentation of the data in Fig. S3 is so confusing that it is difficult to figure out the basis for optimizing the material ratio. It is recommended to make a histogram or a line chart with the ratio of a certain material as the abscissa. In addition, more comparisons should be provided to optimize the proportion of AMPS.
2. As an electrolyte material, ionic conductivity should also be used as a basis for optimizing the material ratio.
3. Higher magnification SEM or TEM images should be provided to illustrate the dispersion of Laponite and GO in hydrogels.
4. The authors indicated in Figure 1 c that the optimal condition for hydrogel self-healing was 808 nm infrared light for 20 min. The conditions used in Fig. 3 c, e, f / Fig. 4d / Fig. S14c are all "808 nm 10 min". Is this contradictory?
5. The article mentions that the presence of GO provides ionic conductivity, please explain.
6. The cyclic reciprocating failure/self-repairing process of the composite hydrogel is carried out at 80°C. If there is no good sealing, the water will easily evaporate at a high temperature of 80°C, which will change the concentration of the electrolyte. Relevant conditions should be described more clearly.
7. The CNT electrode with wrinkled structure is prepared by pre-straining the hydrogel to 900%. How does the CNT film not break under 1000% strain?

Response to the Comments from Reviewer 1:

(1) The authors reported a hydrogel of a copolymer cross-linked by dual linkers of Laponite and graphene oxide. The unique properties and abundant functional groups of cross-linkers provide the resultant hydrogel with high mechanical stretchability, excellent ionic conductivity, and superior self-healable performance under both infrared light and heating treatments. Along with designing wrinkled structure electrodes, supercapacitors fabricated by using this hydrogel as gel electrolyte not only exhibit superior mechanical stretchability of 1000%, but also could achieve highly efficiently repeated self-healable performance with treatments by infrared irradiation and heating. However, this manuscript should be revised in the following items before it can be accepted for publication.

Response: Many thanks for your positive comments and suggestions, which are very helpful to improve our manuscript. According to your suggestion, some improvements have been made in the revised manuscript as following.

(2) The specific capacitance slight decreased with the strain. The author ascribed to the increasing series resistance of devices. However, the contact area between electrode and electrolyte may be changed. The author should give the comprehensive analysis from different reasons.

Response: As suggested, we have discussed it more comprehensively in the Page 8 in the revised manuscript.

(3) The auother should give EIS testing result. For better confirm the good performance of self-healing and stretching, some analysis about the EIS needs to be given.

Response: Thanks for your professional suggestions. Accordingly, more EIS results were given for better understanding the stretching and healing performance of supercapacitors in the revised manuscript. Supplementary Figure 18 was the EIS curves of a supercapacitor being stretched from 0% to 1000% strain, Supplementary Figure 21 was the EIS curves of a supercapacitor healed by irradiation with 808 nm infrared light for 10 min and heating at 80 °C for 60 min after different broken/healed cycles, Supplementary Figure 23 was the EIS curves of a healed supercapacitor being

stretched from 0% to 900%. Relative analysis about the EIS was made in the revised manuscript (Page 8, 10 and 11).

(4) The devices were cut in same position? The author should explain.

Response: Yes, the devices were almost cut in same position, which has been demonstrated in Page 9 of the revised manuscript.

Response to the Comments from Reviewer 2

(1) This manuscript designed a kind of hydrogel electrolyte based on poly(AMPS-co-DMAAm) cross-linked by Laponite and graphene oxide. The resulting electrolyte showed high stretchability and healability. A supercapacitor containing the hydrogel electrolyte not only kept the original electrochemical performances at a high strain of 1000%, but also exhibited desirable healability after multiple mechanical damages. Interestingly, the healed capacitor could also be stretched to a strain of 900% with only 15% capacitance decay. The present investigation offers an alternative strategy to fabricate flexible capacitors with healability and stretchability. However, the following questions should be clarified before the manuscript can be accepted by Nature Communications.

Response: Thanks very much for your positive comments and suggestions, which are very helpful to improve our manuscript. According to your suggestion, some improvements have been made in the revised manuscript as following.

(2) Although the capacitor described by Dr. Chen exhibited healability and stretchability, the similar capacitor and the fabrication strategy had been reported by Chunyi Zhi et al (Nat. Commun. 2015, 6, 10310).

Response: Yes, Zhi et al reported a dual crosslinked polyelectrolyte for self-healing and stretchable supercapacitors, which showed a long healing process (within tens of minutes, non-given a certain time) and a limited stretchability (600%). It did not demonstrate that whether a healed supercapacitor device maintained its stretchability. For comparison, the supercapacitor devices based our new hydrogel electrolyte can be healed within 10 minutes, and showed ultrahigh stretchability (1000%) without obvious change of capacitance. In addition, we for the first time showed that a

broken/healed supercapacitor can achieve a high stretchability of 900% without damaged. The great achievements of this work on the healable and stretchable supercapacitor must greatly promote the development of healable and stretchable electronics.

(3) The healing behavior of the capacitor is attributed to its hydrogel electrolyte. However, the electrodes and current collectors of the capacitor did not show self-healing capability.

Response: Yes, the healing behavior of most reported capacitors is provided by the used hydrogel electrolyte because of the electrodes and current collectors without healing capability. In this work, the broken ends of non-healable CNT-based films were overlapped and pressed to adhere on the hydrogel electrolyte by hand to make them contacted again, and the highly wrinkled structure of CNT-based electrodes (also served as current collectors) provided CNTs electrodes with enough length to overlap with each other. The relative details were demonstrated in the Methods section (Page 13).

(4) The mechanical recovery of the broken hydrogel electrolyte involved external stimulus such as infrared light and heat treatment. In this regard, the present hydrogel electrolyte is only healable instead of self-healable. In addition, the temperature of the spot irradiated by infrared light should be measured. Will the temperature higher than 80 °C?

Response: Thanks for your professional comments and suggestions. According to your comments, “self-healable” were corrected as “healable” in the revised manuscript. As suggested, we have measured the temperature changes of hydrogel and supercapacitors as a spot of infrared light was irradiated. As showed in Supplementary Figure 8 and 20, the temperature of the spot irradiated by infrared light on hydrogel was lower than 80 °C (40.5 °C for 20 min), while the temperature of the spot irradiated by infrared light on supercapacitor was higher than 80 °C within 1 min due to the excellent optical absorbance of carbon nanotubes. Relative discussions were demonstrated in Page 5 in the revised manuscript.

Response to the Comments from Reviewer 3

(1) This manuscript reports new type of poly (AMPS-co-DMAAm)/Laponite/GO nanocomposite hydrogels with ultra-high stretchability and self-healing properties. It is very meaningful for the development of wearable electronic devices and smart electronic textiles. However, some revisions need to be made before its publication.

Response: Thanks very much for your positive comments and suggestions, which are very helpful to improve our manuscript. According to your suggestion, some improvements have been made in the revised manuscript as following.

(2) The presentation of the data in Fig. S3 is so confusing that it is difficult to figure out the basis for optimizing the material ratio. It is recommended to make a histogram or a line chart with the ratio of a certain material as the abscissa. In addition, more comparisons should be provided to optimize the proportion of AMPS.

Response: Many thanks for your professional advices. As suggested, the dependence of mechanically healing efficiencies and ionic conductivities of as-prepared hydrogels on the contents of different components was summarized in Supplementary Figure 5 and 6. As suggested, more comparisons to optimize the proportion of AMPS (2.6% and 9.2%) in hydrogel had been synthesized and measured, the relative data was summarized in Supplementary Figure 5-7.

(3) As an electrolyte material, ionic conductivity should also be used as a basis for optimizing the material ratio.

Response: Thanks for your professional suggestion. Ionic conductivities of as-prepared hydrogels with different ratios among AMPS, DMAAm, Laponite and GO were summarized in Supplementary Figure 7. The optimized ratio of different components was achieved by balancing the mechanical properties, mechanically healing efficiencies and ionic conductivities of as-prepared hydrogels.

(4) Higher magnification SEM or TEM images should be provided to illustrate the dispersion of Laponite and GO in hydrogels.

Response: Thank you for your good suggestion. Higher magnification SEM and TEM images of hydrogels were given in Supplementary Figure 3. From the SEM image, it

can be seen that hydrogel exhibited uniform three-dimensional network structure. The TEM image showed that Laponite was well-dispersed in hydrogel. The relative discussion was added in the second paragraphs in Page 3 in the revised manuscript.

(5) The authors indicated in Figure 1 c that the optimal condition for hydrogel self-healing was 808 nm infrared light for 20 min. The conditions used in Fig. 3 c, e, f / Fig. 4d / Fig. S14c are all "808 nm 10 min". Is this contradictory?

Response: Thanks for your comment on it. It is not contradictory. Due to the used CNTs with excellent optical absorbance and photothermal property³⁴⁻³⁶, the temperature of supercapacitor devices rapidly increased to over 80 °C (Supplementary Figure 20) as the infrared light was irradiated on the surface of CNTs, achieving a much rapider healing process (10 min) than that (20 min) of bare hydrogel. Relative discussion was given in Page 9 in the revised manuscript.

(6) The article mentions that the presence of GO provides ionic conductivity, please explain.

Response: Thanks very much for your comment. Previous research results (*Adv. Funct. Mater.* 2013, **23**, 3353-3360) have been demonstrated that the abundant oxygen-containing function groups in GO could provide efficient channel and facilitate ion transport for supercapacitor applications. Our results (Supplementary Figure 7) also showed that the ionic conductivity of hydrogel increased as some GO (0.1%) was introduced, which were accordant with the previous results.

(7) The cyclic reciprocating failure/self-repairing process of the composite hydrogel is carried out at 80°C. If there is no good sealing, the water will easily evaporate at a high temperature of 80°C, which will change the concentration of the electrolyte. Relevant conditions should be described more clearly.

Response: Many thanks for your professional suggestion. Relevant conditions for cycling failure/repairing measurements of hydrogel and supercapacitors were demonstrated in the Methods section (Page 13).

(8) The CNT electrode with wrinkled structure is prepared by pre-straining the

hydrogel to 900%. How does the CNT film not break under 1000% strain?

Response: Thanks for your professional comments. As showed in Figure 2b and 2c, the formed wrinkled CNT became plane as a strain of 900% was applied. The randomly entangled CNTs in the film were oriented along the stretching direction (Figure 2d, new result), the relative discussion was added in the first paragraph in Page 8 in the revised manuscript.

Reviewers' Comments:

Reviewer #1:

Remarks to the Author:

The authors have addressed our comments and this manuscript can be accepted for publication.

Reviewer #2:

Remarks to the Author:

The authors had revised their manuscript entitled "Ultrastretchable and superior healable supercapacitors based on a novel cross-linked hydrogel electrolyte". The comments of the reviewers had also been replied point by point. However, healable supercapacitors with ultrastretchability had been reported previously by using similar stretchable polymeric electrolytes (Nat. Commun. 2015, 6, 10310). The healing behavior of the present capacitor occurred at 80 °C and involved external stimulus. Compared with the previous reports, on the whole, the present investigation did not show a breakthrough on the design or fabrication of self-healable supercapacitors. Therefore, the novelty and importance of the manuscript will not meet the requirements of Nature Communications at its present form.

Reviewer #3:

Remarks to the Author:

The revised manuscript is greatly improved. I think the manuscript can be accepted.

Dear Reviewers,

Thank you again for your positive and constructive comments on our manuscript. Please feel free to let us know if there is any further question you have. The following is our point-by-point response.

Sincerely yours,

Tao Chen

Response to Reviewer #1:

The authors have addressed our comments and this manuscript can be accepted for publication.

Response: Thank you.

Response to Reviewer #2:

The authors had revised their manuscript entitled “Ultrastretchable and superior healable supercapacitors based on a novel cross-linked hydrogel electrolyte”. The comments of the reviewers had also been replied point by point. However, healable supercapacitors with ultrastretchability had been reported previously by using similar stretchable polymeric electrolytes (Nat. Commun. 2015, 6, 10310). The healing behavior of the present capacitor occurred at 80 oC and involved external stimulus. Compared with the previous reports, on the whole, the present investigation did not show a breakthrough on the design or fabrication of self-healable supercapacitors. Therefore, the novelty and importance of the manuscript will not meet the requirements of Nature Communications at its present form.

Response: Thanks for your comments. Previously, the authors (Nat. Commun. 2015, 6, 10310) reported a polyelectrolyte for self-healing and stretchable supercapacitors, which showed a long healing process (within tens of minutes, non-given a certain time) and a limited stretchability (600%). However, it did not demonstrate that whether a healed supercapacitor device maintained its stretchability. For comparison, the supercapacitor devices based our hydrogel electrolyte can be healed within 10 minutes, and showed ultrahigh stretchability (1000%). In addition, a broken/healed supercapacitor can maintain a high stretchability of 900% without damaged, which is demonstrated for the first time. Therefore, our new report has obvious breakthrough on the stretchable and healable performance of supercapacitors.

Response to Reviewer #3:

The revised manuscript is greatly improved. I think the manuscript can be accepted.

Response: Thank you.